# Entanglement between more than two hundred macroscopic atomic ensembles in a solid

P. Zarkeshian[1], C. Deshmukh [1], N. Sinclair[1], S.K. Goyal[1], G.H. Aguilar[1], P. Lefebvre[1], M.Grimau Puigibert[1], V.B. Verma[2], F. Marsili[3], M.D. Shaw[3], S.W. Nam[2], K. Heshami [4], D. Oblak[1], W. Tittel[1] & C. Simon[1]

There are both fundamental and practical motivations for studying whether quantum entanglement can exist in macroscopic systems. However, multiparty entanglement is generally fragile and difficult to quantify. Dicke states are multiparty entangled states where a single excitation is delocalized over many systems. Building on previous work on quantum memories for photons, we create a Dicke state in a solid by storing a single photon in a crystal that contains many large atomic ensembles with distinct resonance frequencies. The photon is re-emitted at a well-defined time due to an interference effect analogous to multi-slit diffraction. We derive a lower bound for the number of entangled ensembles based on the contrast of the interference and the single-photon character of the input, and we experimentally demonstrate entanglement between over two hundred ensembles, each containing a billion atoms. We also illustrate the fact that each individual ensemble contains further entanglement.

[1] Institute for Quantum Science and Technology, and Department of Physics & Astronomy, University of Calgary, 2500 University Drive NW, Calgary, AB, Canada T2N 1N4. [2] National Institute of Standards and Technology, Boulder, CO 80305, USA. [3] Jet Propulsion Laboratory, California Institute of Technology, 4800 Oak Grove Drive, Pasadena, CA 91109, USA. [4] National Research Council of Canada, 100 Sussex Drive, Ottawa, ON, Canada K1A 0R6. Correspondence and requests for materials should be addressed to C.S. (email: christoph.simon@gmail.com)

The question whether quantum superposition and entanglement can exist in macroscopic systems has been at the heart of foundational debates since the beginnings of quantum theory[1–5] and inspired many experiments[6–16]. One important type of entangled states corresponds to a single excitation that is delocalized over many systems. Such Dicke states[17] have been created for individual photons[18, 19] and cold atoms[20–22], reaching up to three thousand atoms[23]. In solids, superradiance associated with a Dicke state was recently demonstrated for two superconducting qubits[24].

It is well known that, in the ideal case, a Dicke state is created whenever a single photon is stored in an atomic ensemble[25]. However, in real experiments, neither the initial single-photon state nor the quantum storage process is perfect. It is not a priori obvious whether the multiparty entanglement will survive under these conditions. Atomic-ensemble-based quantum memories have been studied intensively both in atomic gases[26–32] and in solid-state systems[33–37]. One widely used quantum storage method is the atomic frequency comb (AFC) quantum memory[38–41]. An AFC is a collection of atomic ensembles with different, equally spaced resonance frequencies. Such AFCs can be conveniently generated in rare earth ion-doped crystals through optical pumping. Each individual ensemble represents one "tooth" of the comb.

Here we create entanglement between these teeth by the absorption of a single photon. More precisely, there is a small probability of absorbing more than one photon, and it is essential to take this into account when attempting to quantify the multiparty entanglement. We derive a criterion that allows us to show that multiparty entanglement of over two hundred ensembles is present for our experimental conditions.

## Results

For most of this work, we focus on the entanglement that is generated between the teeth, rather than within each tooth. It is then possible to treat each tooth as a single two-level system (a "qubit") with collective states $|0\rangle$ (where all atoms in the tooth are in the ground state) and $|1\rangle$ (where a single atom in the tooth is excited). Ideally, the absorption of a single photon by an AFC consisting of $N$ teeth creates the lowest-order Dicke state, widely known as the W state,

$$|W\rangle_N = \frac{1}{\sqrt{N}}(|100...0\rangle + |010...0\rangle... + |000...1\rangle), \quad (1)$$

where the first term corresponds to the case in which the first tooth has absorbed the photon, etc. Our theoretical and experimental approach to demonstrating this multiparty entanglement is based on the re-emission of the single photon from the AFC, which is due to a collective interference effect. As time passes, the different terms in the above equation acquire different phases $e^{i\delta_j t}$, where $\delta_j = j\Delta$ stands for the detuning of the $j$th tooth relative to the lowest-frequency ($j = 0$) tooth, $j$ runs from 0 to $N-1$, and $\Delta$ represents the angular frequency spacing between the teeth. The photon has a high probability of being re-emitted only at the "echo" times when all the phase factors are the same, i.e., at times that are integer multiples of $2\pi/\Delta$[38, 39]. This echo emission is thus an interference effect in time that is very similar to spatial multislit diffraction (see Fig. 1). This analogy is at the heart of our approach, which we now describe in detail.

**Derivation of a lower bound for the entanglement depth.** Our method for demonstrating the entanglement between the teeth is

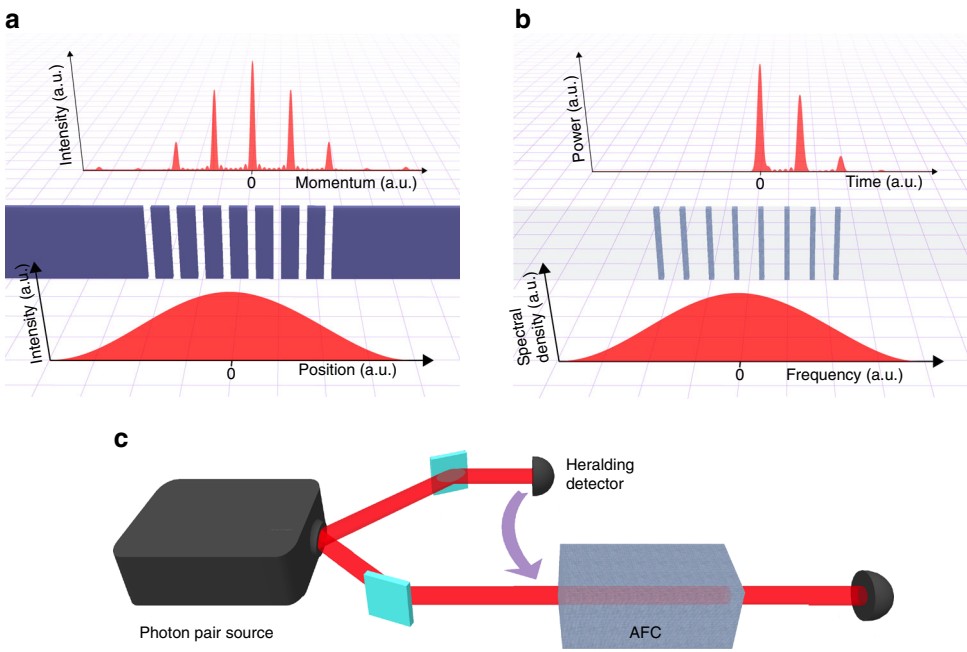

**Fig. 1** Principle of our approach. Our approach is based on the analogy between re-emission of a photon from an atomic frequency comb and multi-slit diffraction. **a** Light passing through a mask with a spatially periodic structure, i.e., a diffraction grating, travels different optical path lengths depending on the part of the grating through which it was transmitted. This results in sharp constructive interference and broad destructive interference in momentum space. **b** Light directed into an atomic ensemble with a spectrally periodic absorption profile (AFC) is absorbed in these atoms, causing them to oscillate at their (different) resonant frequencies. This results in an interference in time, manifested via sharp peaks in the re-emission probability at well-defined times ("echos"). The absorption "teeth" in the AFC are analogous to the slits in the diffraction grating. **c** Principle of the experiment. A single photon is created with the help of a photon pair source and a heralding detector. The single photon is stored in an AFC and detected after retrieval from the AFC. The echo contrast in combination with the single-photon character of the source can be used to find a bound on the minimum number of entangled teeth

inspired by the fact that, for multi-slit interference, the number of participating slits can be inferred from the sharpness of the interference pattern. We consider the ratio of the maximum photon re-emission probability in the first echo to the re-emission probability averaged over one period $2\pi/\Delta$,

$$R = \frac{P\left(\frac{2\pi}{\Delta}\right)}{\frac{\Delta}{2\pi}\int_{\frac{\pi}{\Delta}}^{\frac{3\pi}{\Delta}} P(t)\mathrm{d}t}. \quad (2)$$

We refer to $R$ as the "echo contrast". The photon emission probability $P(t)$ is proportional to $\langle S_+(t)S_-(t)\rangle$[39, 42], where $S_-(t) = \sum_j e^{i\delta_j t} S_-^j$ and $S_-^j = |0\rangle^j\langle 1|^j$ is the dipole operator for tooth $j$, and $S_+(t) = S_-^\dagger(t)$. Here we are using the Heisenberg picture, where observables rather than states are time dependent. At the time of the first echo, $t_e = 2\pi/\Delta$, all teeth are in phase, giving $S_-(t_e) = \sum_j S_-^j \equiv S_-$. On the other hand, averaging the re-emission probability over a time interval $2\pi/\Delta$ centered at the echo time as in Eq. (2) leads to $\frac{\Delta}{2\pi}\int_{\frac{\pi}{\Delta}}^{\frac{3\pi}{\Delta}} \mathrm{d}t \left\langle \sum_{j,l} S_+^j S_-^l e^{i(l-j)t} \right\rangle$, which is non-zero only for $j = l$. Then, the denominator of Eq. (2) results

in the expression $\left\langle \sum_j |1\rangle^j\langle 1|^j \right\rangle$, i.e., the sum of the excitation probabilities for each tooth, yielding $R = \frac{\langle S_+ S_-\rangle}{\left\langle \sum_j |1\rangle^j\langle 1|^j \right\rangle}$.

We now show that the echo contrast $R$ is closely related to the "entanglement depth"[43]. A (generally mixed) quantum state of $N$ qubits has entanglement depth at least equal to $M$ if it cannot be decomposed into a convex sum of product states with all factors involving less than $M$ entangled qubits, i.e., at least one of the terms needs to be an $M$-qubit entangled state.

First, we consider the case where exactly one photon is absorbed by the AFC. In this case, $R = \langle S_+ S_-\rangle$. Let us suppose that $R$ is found experimentally to have a value of $M$. This value can be achieved by the state $|W\rangle_M \otimes |0\rangle^{\otimes(N-M)}$, i.e., a Dicke state involving $M$-teeth and no excitation in the remaining $N-M$-teeth. Let us note that $S_+ S_-$ is permutation invariant (as is $R$ in the general case), so all permutations of the teeth are equivalent for our purpose. The above state has an entanglement depth of $M$. All other states in the single-excitation subspace giving $R = M$ involve more than $M$ entangled teeth, i.e., they are of the form $\sum_{j=1}^N c_j |0...1_j...0\rangle$ with more than $M$ non-zero coefficients $c_j$ that

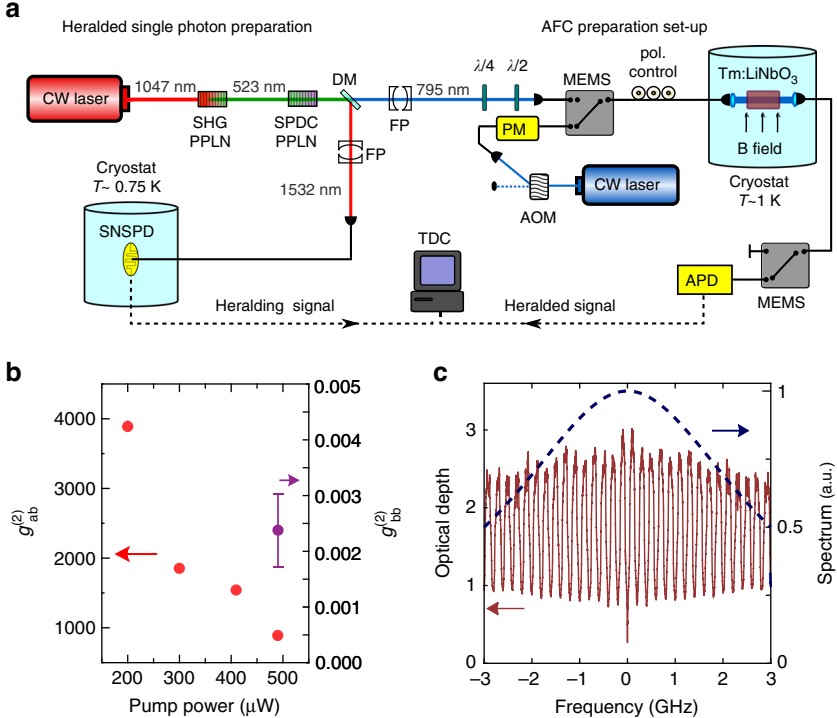

**Fig. 2** Experimental set-up. **a** *Left*: Source of heralded single photons. A 1047 nm continuous wave (CW) laser is frequency doubled and subsequently downconverted with two different periodically poled lithium niobate (PPLN) nonlinear crystals. This probabilistically generates photon pairs with central wavelengths at 795 and 1532 nm, which are separated using a dichroic mirror (DM). A Fabry–Perot (FP) cavity spectrally filters the 795 nm photons to 6 GHz in order to match the spectral acceptance bandwidth of our AFC. Wave plates allow adjusting the polarization of the photons to maximize interaction with the AFC. Similarly, the 1532 nm photons are filtered down to 10 GHz using another FP cavity before being detected by a superconducting nanowire single-photon detector (SNSPD)—this detection heralds the presence of the 795 nm photon. *Right*: AFC preparation. CW laser light at 795 nm, which is frequency and amplitude modulated by an acousto-optic modulator (AOM) and a phase modulator (PM), is used to optically pump the Tm atoms in a cryogenically cooled Tm:LiNbO₃ bulk crystal to create an AFC. Micro electro-mechanical switches (MEMS) allow switching between optical pumping and storage of single photons. An avalanche photodiode (APD) is used to detect the 795 nm photons after interacting with the AFC. The electrical signal from both detectors (APD and SNSPD) is analyzed using a time-to-digital converter (TDC) to register the difference in arrival times of the two detection signals. **b** Characterization of the single-photon source. Cross-correlation function $g_{ab}^{(2)}$ of the two downconverted photons at 1532 nm (mode a) and 795 nm (mode b) as a function of the pump power. The *error bar* indicates the standard deviation derived from the Poissonian statistics of the photon detection events. We also show the heralded autocorrelation function of the 795 nm photon $g_{bb}^{(2)}$ for one value of the pump power. **c** A sample trace of an AFC (*solid line*) with $N$ = 30 teeth and bandwidth $B$ = 6 GHz and the spectral-density profile of the 795 nm photon (*dashed line*)

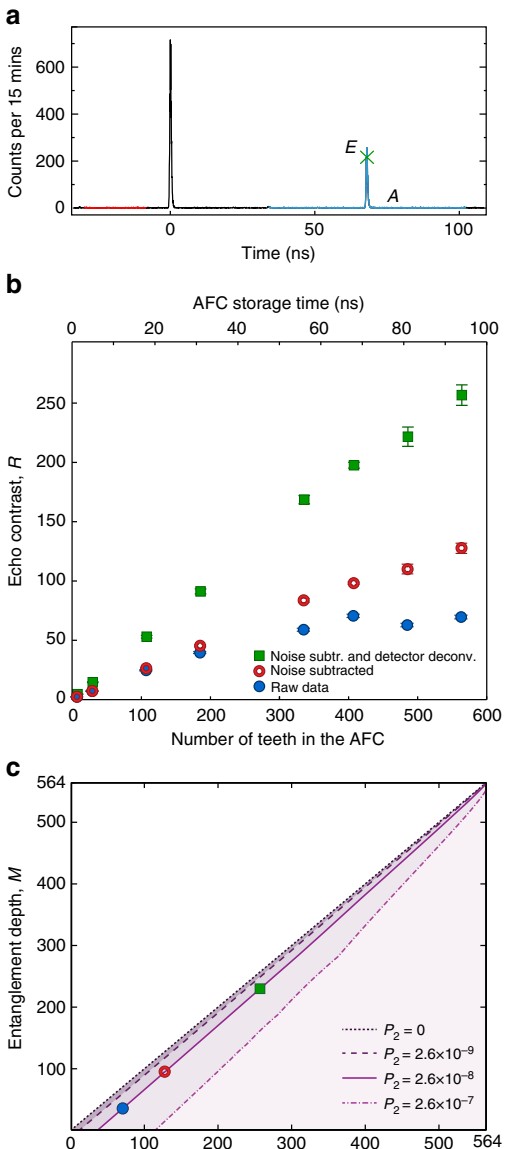

**Fig. 3** Measured echo contrast and bounds for entanglement depth. **a** An example of a time histogram for a storage time of 68 ns. The peak value of the echo ($E$), marked by the *green x*, is extracted from fitting the echo to a Gaussian profile, which reduces the impact of TDC sampling noise, and the *blue region* around the echo represents the time interval over which counts are averaged ($A$). The *red region*, which precedes both the transmitted and echo photon, is used to extract the background noise level. **b** Experimental values of echo contrast $R$ as a function of the number of teeth $N$. *Blue circles* represent the values of $R$ obtained from raw data, *red open circles* are obtained after noise subtraction, and *green squares* after noise subtraction and deconvolution of the detector response (see "Methods" section). *Error bars* indicate standard deviations derived from the Gaussian fitting and Poissonian statistics of the photon detection events. **c** Numerical bounds for the entanglement depth $M$ as a function of echo contrast $R$ for the experimental values $P_1 = 3.5 \times 10^{-3}$ for the single-excitation probability, $N = 564$ for the number of teeth, and $P_2 = 2.6 \times 10^{-8}$ for the double-excitation probability (*solid line*). For comparison, we also show the bounds for the same $P_1$ and $N$, but for $P_2 = 0$ (*short-dashed line*), $P_2 = 2.6 \times 10^{-9}$ (*dashed line*), and $P_2 = 2 \times 10^{-7}$ (*dash-dotted line*). The experimental values of the echo contrast $R$ are shown as a *blue dot* (raw data), *red dot* (after noise subtraction), and *green square* (after noise subtraction and detector deconvolution) respectively

are not all equal and fulfilling $\langle S_+ S_- \rangle = \left| \sum_{j=1}^{N} c_j \right|^2 = M$. This is due to the fact that, in the single-excitation subspace of the $M$-teeth Hilbert space, the Dicke state $|W\rangle_M$ is the only eigenstate of $S_+ S_-$ with a non-zero eigenvalue (namely $M$), see "Methods" section. Thus an experimental value $R = M$ implies an entanglement depth of at least $M$ in the case of perfect absorption of a single photon.

Next, we consider the more realistic case of absorption of a single photon with probability $P_1 < 1$. In this case, $R = \langle S_+ S_- \rangle / P_1$. Let us again suppose that the experiment gives $R = M$. This can be achieved with a mixed state or coherent superposition state that has a probability weight $P_1$ for $|W\rangle_M \otimes |0\rangle^{\otimes(N-M)}$ and a weight $1 - P_1$ for $|0\rangle^{\otimes N}$. With the same line of arguments as for the case of the perfect absorption ($P_1 = 1$), the entanglement depth is again at least $M$.

The situation becomes more complex if there is a chance for the AFC to absorb more than one photon. In our experiment, the most significant higher-order probability is $P_2$, the probability of absorbing two photons, which is very small but non-zero. We now have to consider states with higher-order components, e.g., the fully separable state $(\alpha|0\rangle + \beta|1\rangle)^{\otimes N}$ with $\alpha^2 + \beta^2 = 1$. Since this state has the Dicke state of $N$ teeth, $|W\rangle_N$, as its component in the single-excitation subspace, it can give very large values of the echo contrast up to $R \simeq N$, which happens in the limit of small $\beta$. On the other hand, in this limit, the two-excitation probability is $P_2 \simeq P_1^2 / 2$. In our experiment, the values of $P_1$ and $P_2$ satisfy $P_2 \ll P_1^2 / 2$, see below. It is clear from this example that the values of $P_1$ and $P_2$ are important for deciding whether large values of $R$ imply large values of the entanglement depth. We have numerically determined a lower bound for the entanglement depth, $M$, as a function of the echo contrast, $R$, conditioned on the experimental values of $P_1$ and $P_2$ (see "Methods" section). This bound can be very well approximated in our regime by the relation:

$$ M > R - \frac{\sqrt{2P_2}}{P_1} N. \qquad (3) $$

Note that the bound gives $M > 0$ (which is consistent with the correct value $M = 1$) for the fully separable state discussed above, for which $R = N$, but also $\frac{\sqrt{2P_2}}{P_1} = 1$. Below we utilize the obtained numerical bound to find the minimum entanglement depth corresponding to the experimental value of the echo contrast.

**Experimental results**. The principle of our experimental approach is shown in Fig. 1c, and the detailed experimental set-up is shown in Fig. 2a. It consists of a source of heralded single photons, an AFC, and a time-resolved detector to register the retrieved photons. The heralded single-photon source is implemented by spontaneous parametric down conversion (SPDC), which probabilistically down converts photons at 523.5 nm into pairs of photons with wavelengths centered around 795 and 1532 nm. Provided a single pair was generated, detection of a 1532 nm photon heralds the presence of a 795 nm photon. Measuring the cross-correlation function of the source, shown in Fig. 2b, allows us to determine the mean photon number per mode $\mu$, where the mode is defined by the coherence time of the pump laser. We find $\mu = (1.1 \pm 0.1) \times 10^{-3}$ (see "Methods" section). This low mean photon number indicates that the probability of generating multi pairs is very low (of the order $10^{-6}$ per mode), thus confirming the nearly single-photon nature of our heralded source. This can also be seen from the heralded autocorrelation function in Fig. 2b. In combination with a precise estimation of the loss at various places of the set-up and the absorption efficiency in the AFC (see

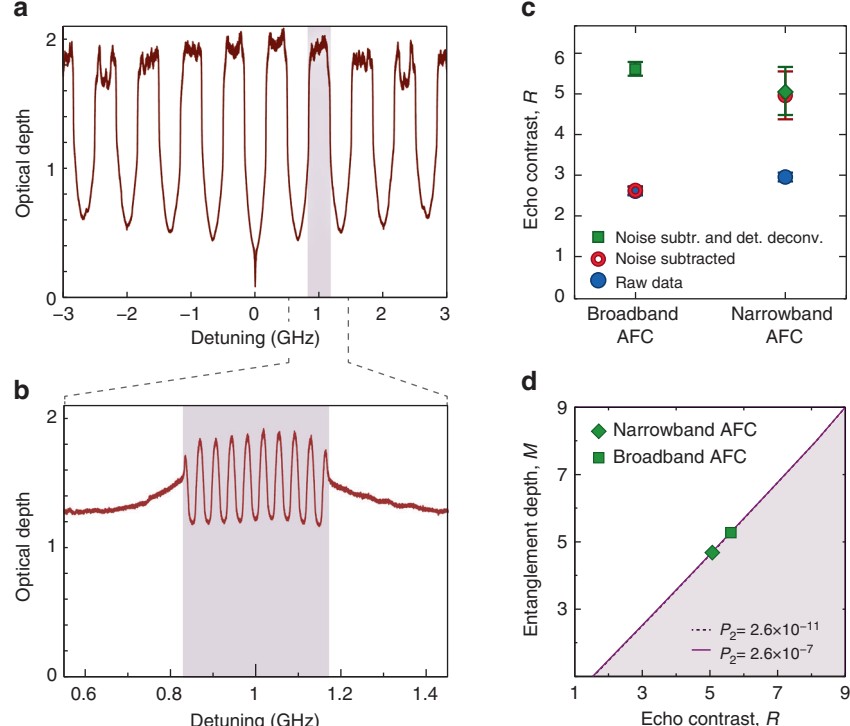

**Fig. 4** Entangled subsystems within each ensemble. **a** An AFC with a bandwidth of 6 GHz with $N = 9$ teeth, where each tooth is 0.33 GHz wide. **b** The plot shows an AFC with a narrower bandwidth of 0.33 GHz that is nested within a single tooth of the broadband AFC. This AFC is also created with $N = 9$ teeth. **c** Echo contrast for the broadband and narrowband AFCs. **d** Bounds on the entanglement depth as a function of echo contrast for the experimental values of $P_1 = 1.1 \times 10^{-2}$ and $P_2 = 2.4 \times 10^{-7}$ for the broadband AFC with $N = 9$, and $P_1 = 8.8 \times 10^{-5}$ and $P_2 = 1.6 \times 10^{-11}$ for the narrowband AFC with $N = 9$. It can be inferred from these two plots that the entanglement depth is at least 5 for the broadband and at least 4 for the narrowband AFC. *Error bars in the last two plots indicate standard deviations derived from the Gaussian fitting and Poissonian statistics of the photon detection events*

"Methods" section), this allows us to determine $P_1$ and $P_2$ using a detailed theoretical model of the set-up (see "Methods" section). We find $P_1 = (3.50 \pm 0.03) \times 10^{-3}$ and $P_2 = (2.55 \pm 0.23) \times 10^{-8}$, where the uncertainties in $P_1$ and $P_2$ are dominated by the uncertainties in the optical depth of the AFC and in the mean photon number per mode, respectively.

The AFC for the 795 nm photons is implemented in a bulk Tm: LiNbO$_3$ crystal. Optical pumping is used to spectrally shape the inhomogeneously broadened $^3H_6 \rightarrow \, ^3H_4$ absorption line of the Tm atoms into a series of absorption peaks (teeth) spaced by angular frequency $\Delta$, see Fig. 2c. The number of teeth $N$ in the AFC is given by $N = B*2\pi/\Delta$, where $B$ is the bandwidth of the AFC (fixed at 6 GHz); and $N$ can be changed by modifying $\Delta$. After a storage time of $t_e = 2\pi/\Delta$, the 795 nm photons are retrieved and detected by a single photon detector whose signal is sent to a time-to-digital converter (TDC). This generates a time histogram of the 795 nm photon detections such as the one in Fig. 3a.

The echo contrast $R$ can be directly extracted from the histogram by taking the ratio of the peak value $E$ of the echo to the average of the counts $A$ in a time interval centered on the echo and equal to the storage time, i.e., $R = E/A$. Figure 3b shows a plot of the ratio $R$ as a function of the number of teeth $N$ in the AFC. We see that the value of $R$ obtained from the raw data (*solid blue circles*) first increases with $N$, reaches a maximum of $70.6 \pm 1.4$ for $N = 408$, and then saturates. The main factors that limit the value of $R$ are on the one hand the limited retrieval efficiency of the AFC and detector jitter, both of which limit the peak value $E$ reached by the echo, and on the other hand noise coming from multi-pair emissions of the source and detector dark counts, both of which increase the denominator $A$ of the echo contrast $R$. After

noise subtraction (*red circles*) and additional deconvolution of the finite detector response time (*solid green squares*), we find a maximum $R$ of $127.6 \pm 4.3$ and $256.7 \pm 8.7$, respectively, both for $N = 564$. The linearity of the echo contrast with $N$, after compensating for the above effects (see "Methods" section for details), is consistent with the expectation, since greater values of $N$ correspond to larger Dicke states.

In Fig. 3c, we plot our numerical bounds for the entanglement depth $M$ as a function of echo contrast $R$ for $N = 564$ and taking into account the experimental values of $P_1 = 3.5 \times 10^{-3}$ and

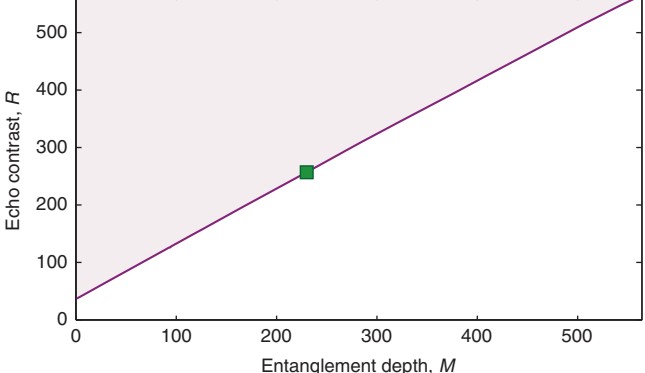

**Fig. 5** Bound for maximum value of echo contrast $R$ as a function of entanglement depth $M$. We assume $P_1 = 3.5 \times 10^{-3}$ and $P_2 = 2.6 \times 10^{-8}$ for the single-excitation and double-excitation probability, respectively, and the total number of ensembles (teeth) $N = 564$. The experimental value of the echo contrast $R$ is shown as a *green square*

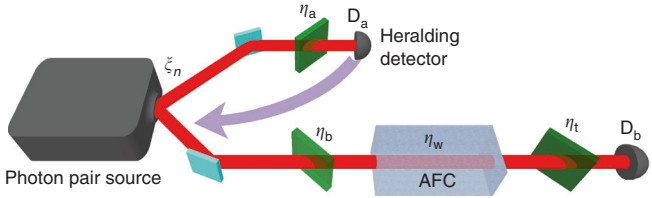

**Fig. 6** Visualization of our mathematical model for the set-up. This figure visualizes our mathematical model of the set-up of Fig. 1c; $\xi_n$ is the amplitude for creating $n$ pairs, $\eta_a$ is the overall transmission and detection probability in mode a, $D_a$ is the heralding detector, $\eta_b$ is the transmission before the memory in mode b, $\eta_w$ is the write efficiency of the atomic frequency comb memory, $\eta_t$ is the transmission and detection probability after the memory in mode b, and $D_b$ is the detector in mode b. All losses and inefficiencies are modeled as beam splitters

$P_2 = 2.6 \times 10^{-8}$, see also Eq. (3). We include other values of $P_2$ for comparison. (Note that $P_2 \ll P_1$ in all cases, so $P_1$ is kept essentially constant.) We conclude that the noise-and-jitter-corrected echo contrast of $R = 256.7 \pm 8.7$ implies at least $229 \pm 11$ entangled teeth, where the uncertainty in the entanglement depth is dominated by the uncertainty in $R$.

Each of the entangled teeth above consists of many atoms (of the order $10^9$, see "Methods" section). As a consequence, each state $|1\rangle$ in Eq. (1) is a type of Dicke state itself. To illustrate this, we first create an AFC with $N = 9$ (red trace in Fig. 4a) and extract a minimum entanglement depth of 5 from the data. Using only the atoms that form one of the teeth in the first AFC, we then create a second—nested—AFC, again with $N = 9$ (see Fig. 4b). We experimentally determine the minimum entanglement depth for this secondary AFC to be 4, see Fig. 4c, d. This procedure of subdividing an ensemble, while restricted by the laser linewidth in our experiment, is fundamentally only limited by the homogeneous linewidth of the atoms. In principle, it would be possible to demonstrate entanglement of a very large number of entangled subsystems (and even subsubsystems, etc.) in this way.

## Discussion

The remaining mismatch between the obtained values of $R$ and the maximum possible value $N$ can be explained by other experimental limitations, such as the imperfect creation of the AFC and the Lorentzian spectral-density profile of the photons, which limit the contribution to the interference coming from the teeth that are further detuned. The state that is actually generated in the experiment is thus of the form $\sum_{j=1}^N c_j |0\ldots01_j0\ldots0\rangle$ (not all $c_j$ being equal) rather than a perfect W state. However, this only means that our bound on the entanglement depth is conservative, because a state with unequal coefficients has to involve a greater number of entangled teeth in order to achieve the same value of $R$. The most relevant type of decoherence in our experiment is irreversible dephasing due to the finite spectral width of each tooth. This effect accumulates over time and thus affects the observed value of $R$. The entangled state at the time of absorption is therefore likely to have had a higher entanglement depth than what can be inferred from observing the echo, which is emitted some time later.

Besides its fundamental interest, the multi-partite entanglement that we have demonstrated may be useful for quantum metrology[23], provided that the storage efficiency can be increased substantially, which is possible, e.g., using cavities[35]. The AFC system allows one to address individual or sets of teeth in frequency space, which in principle makes it possible to characterize the created quantum state in more detail. The large ratio of

inhomogeneous to homogeneous linewidths in rare-earth-doped crystals suggests that it should be possible to create entanglement between larger numbers of teeth, possibly up to one hundred million. The length and the doping concentration of rare earth materials can also be increased, allowing one to additionally increase the number of atoms in each tooth. Storing entangled photons in AFCs created in separated crystals would offer the possibility to study the nature of multi-partite entanglement at yet a higher nesting level. Furthermore the present approach in principle allows the storage of more than one photon, which would enable the creation of more complex entangled states such as higher-order Dicke states.

We note that entanglement between many individual atoms in a solid is demonstrated in a parallel submission by F. Fröwis et al., using a related, but complementary approach based on the directionality of the echo emission from an atomic frequency comb.

## Methods

**Only the Dicke state has a non-zero eigenvalue**. Here we prove the statement that in the single-excitation subspace of the $M$-teeth Hilbert space the Dicke state $|W\rangle_M$ is the only state with a non-zero eigenvalue. For this purpose, it is useful to view each qubit as a spin-1/2 system, such that the dipole operators $|1\rangle\langle0|$ and $|0\rangle\langle1|$ are now viewed as spin raising and lowering operators, respectively. In this spin representation, the zero-excitation state $|0\ldots0\rangle$ has total spin $M/2$ and $S_z$ projection $-M/2$, and $|W\rangle_M$ (which is also completely symmetric and thus belongs to the irreducible representation with maximum total spin) has total spin $M/2$ as well, but $S_z$ projection $-M/2 + 1$. In addition to $|W\rangle_M$, there are $M - 1$ other basis states spanning the single-excitation subspace. They all have the same $S_z$ projection as the Dicke state (i.e., $S_z = -M/2 + 1$), but total spin $M/2 - 1$ since they are not fully symmetric. They have the lowest possible $S_z$ projection value that is compatible with their total spin value, and for that reason, they are annihilated by the spin lowering operator $S_-$. Hence, $|W\rangle_M$ is the only eigenstate of $S_+S_-$ in the single-excitation subspace that has a non-zero eigenvalue.

**Deriving bounds on the entanglement depth**. In order to derive lower bounds on the entanglement depth $M$ for given $R$, $P_1$ and $P_2$, it is convenient to—equivalently—derive upper bounds on $R$ for given $M$, $P_1$ and $P_2$. The fully separable example discussed in the text shows that $R$ can be much greater than the entanglement depth $M$ once $P_2$ is different from zero. For a given $M$, we are allowed to use entangled states of (up to) $M$-teeth as the individual factors, where the factors $\alpha|0\rangle + \beta|1\rangle$ in the fully separable example correspond to $M = 1$. Keeping in mind that $P_1, P_2 \ll 1$ in our experiment, it is clear that there should be a significant vacuum component $|0\rangle^{\otimes M}$ in each factor. As for the non-vacuum part, the Dicke state $|W\rangle_M$ is clearly a good choice because it is the single-excitation state that maximizes $R = \langle S_+S_-\rangle / \langle \sum_j |1\rangle^j\langle1|^j\rangle$ in each $M$-teeth subspace. In fact, it is the optimum choice. Higher-order Dicke states have greater $\langle S_+S_-\rangle$, but not $R$, because they also have greater values for the total number of excitations $\langle \sum_j |1\rangle^j\langle1|^j\rangle$. Moreover—in contrast to $|W\rangle_M$—they require a non-zero $P_2$ (alternatively stated, they make a non-zero contribution to $P_2$), whose value is one of our constraints. Higher-order non-Dicke states are clearly suboptimal. We conclude that the optimal state should contain factors of the form $\alpha|0\rangle^{\otimes M} + \beta|W\rangle_M$. However, it is not immediately apparent how many such factors there should be. For example, consider the states $|\psi_1\rangle = (\alpha_1|0\rangle^{\otimes M} + \beta_1|W\rangle_M) \otimes |0\rangle^{\otimes N-M}$ and $|\psi_2\rangle = (\alpha_2|0\rangle^{\otimes M} + \beta_2|W\rangle_M)^{\otimes2} \otimes |0\rangle^{\otimes N-2M}$. The latter state has a component $|W\rangle_{2M}$ in the single-excitation subspace and will thus achieve a greater value of $R$ compared to $|\psi_1\rangle$. But it also makes a contribution to $P_2$ because of the cross term that is proportional to $\beta_2^2$, whereas $|\psi_1\rangle$ makes no such contribution. This suggests that in general, the optimal state may be a mixed state of the form:

$$\rho = \sum_i q_i |\psi_i\rangle\langle\psi_i|, \qquad (4)$$

with $\sum q_i = 1$, $q_i \geq 0$, and

$$|\psi_i\rangle = (\alpha_i|0\rangle^{\otimes M} + \beta_i|W\rangle_M)^{\otimes i} \otimes |0\rangle^{\otimes(N-iM)}, \qquad (5)$$

where $1 \leq i \leq k$, such that $k = \lfloor N/M \rfloor$. Let us first consider the case where $N$ is divisible by $M$, in which case one simply has $k = N/M$. Note that it is optimal for the coefficients of the (non-vacuum) factors in each $|\psi_i\rangle$ to be identical, in order to maximize the overlap with the state $|W\rangle_{iM}$ in the single-excitation subspace and thus maximize $R$. The ratio $R$ and the probabilities $P_1$ and $P_2$ for the state $\rho$ are

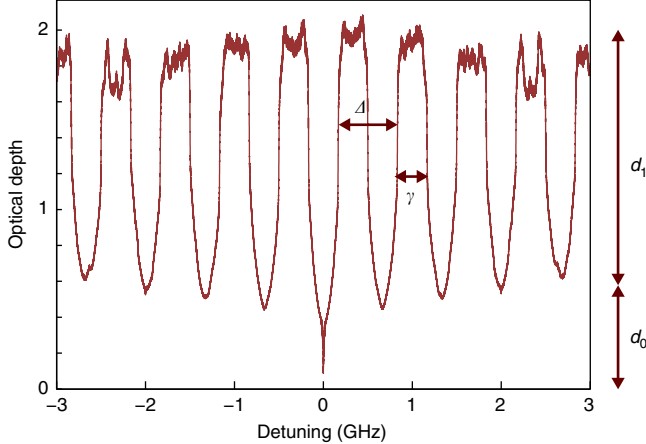

**Fig. 7** A sample trace of an atomic frequency comb. This comb has $N = 9$ teeth and and a bandwidth $B = 6$ GHz. The sample is derived from an average over traces of 50 experimental runs; $d_1$ indicates the peak-to-peak optical depth that constitutes the AFC while $d_0$ is the background optical depth. The finesse $F$ of the AFC is given by the ratio $\Delta/\gamma$, where $\Delta$ is the separation between the teeth and $\gamma$ is the linewidth of the teeth

given by:

$$R = \frac{M}{P_1 + 2P_2} \sum_{n=1}^{k} n^2 q_n \left(\beta_n^2\right) \left(1 - \beta_n^2\right)^{n-1}, \quad (6)$$

$$P_1 = q_1 \beta_1^2 + \sum_{n=2}^{k} n q_n \left(\beta_n^2\right) \left(1 - \beta_n^2\right)^{n-1}, \quad (7)$$

$$P_2 = \sum_{n=2}^{k} \frac{n(n-1)}{2} q_n \left(\beta_n^4\right) \left(1 - \beta_n^2\right)^{n-2}. \quad (8)$$

To derive the upper bounds on $R$, we maximize Eq. (6) with the constraints of Eqs. (7) and (8), where the values of $P_1$ and $P_2$ are obtained from our experiment. We use a global numerical search algorithm that runs a local nonlinear programming solver, which finds the maximum value of a constrained nonlinear multivariable function for multiple start points. It finally reports the global maximum value of $R$ for a given $M$.

The numerical maximization shows that only the $q_1$ and $q_k$ components in Eq. (4) are non-zero in the optimal case, and the latter is close to one. This can be understood physically. The state $|\psi_k\rangle$ has the largest possible number of non-separable states of size $M$, giving a state $|W\rangle_N$ in the single-excitation subspace, which maximizes $\langle S_+ S_-\rangle$. Thus, the weight of this state, $q_k$, should be as large as possible, and indeed it comes out close to 1 in the maximization. However, the size of $\beta_k$ is constrained by the value of $P_2$, which is very small in our case. As a consequence, this state only makes a small contribution to $P_1$. The remaining contribution to $P_1$ is best provided by a state that does not increase $P_2$, i.e., $|\psi_1\rangle$.

In general, $N$ is not always divisible by $M$ and one has $N = kM + k'$, such that $k' < M$ (while $k = \lfloor N/M \rfloor$). In this case, the same arguments apply with the small difference that $|\psi_k\rangle = \left(\alpha_k|0\rangle^{\otimes M} + \beta_k|W\rangle_M\right)^{\otimes k} \otimes \left(\alpha_{k'}|0\rangle^{\otimes k'} + \beta_{k'}|W\rangle_{k'}\right)$. The optimum values for $\alpha_{k'}$ and $\beta_{k'}$ are such that the state $|W\rangle_N$ is obtained in the single-excitation subspace.

Figure 5 shows the maximum possible contrast, $R$, as a function of entanglement depth, $M$, constrained on our experimental values of $P_1$ and $P_2$. In this figure, the curve is close to a straight line. This can be explained by noting that only $q_1$ and $q_k$ are non-zero ($q_1 + q_k \simeq 1$), and $P_2 = \frac{1}{2} k^2 q_k \beta_k^4$ is very small, which makes the contribution of $q_k$ to $P_1$ negligible ($k\beta_k^2 \ll P_1$ for any value of $q_k$) resulting in $P_1 \simeq q_1 \beta_1^2$. Thus, one can obtain $R = \frac{1}{P_1 + 2P_2} \left(MP_1 + \sqrt{2(1 - q_1)P_2 N}\right)$, which gives $R_{\max} \simeq M + \frac{\sqrt{2P_2}}{P_1} N$ and thereby leads to Eq. (3).

**Theoretical model.** The atomic state of the AFC system after absorption depends on the photon statistics of the source, on transmission loss, and on inefficiencies such as inefficient single-photon heralding and absorption in the AFC. We now describe a theoretical model that allows extracting $P_1$ and $P_2$ from experimental data.

In our experiment, an intense laser beam passes through a spontaneous parametric down conversion crystal (SPDC) that converts photons into entangled

pairs of photons traveling in different spatial modes a and b (see Fig. 6). A non-number-resolving photon detector $D_a$, henceforth referred to as the heralding detector, is placed in mode a, and the AFC system followed by another detector $D_b$ is located in mode b. A detection by the detector $D_a$ heralds the presence of at least one photon in mode b.

The combined state of the two downconverted photons in modes a and b can be written as:

$$|\psi\rangle = \sum_{n=0}^{\infty} \frac{\xi_n}{n!} \left(a^\dagger\right)^n \left(b^\dagger\right)^n |0\rangle_a |0\rangle_b, \quad (9)$$

where

$$|\xi_n\rangle^2 = \frac{\mu^n}{(\mu + 1)^{n+1}} \quad (10)$$

is the thermal photon number distribution. Here $a^\dagger$ ($b^\dagger$) is the photon creation operator in mode a (b), $|\xi_n|^2$ is the probability of creating $n$ pairs of photons (one photon per pair in mode a and one in b), and $\mu$ is the mean photon pair number per temporal mode (per coherence time of the pump laser). Let us note that our source is in fact slightly temporally multi mode, which implies that the distribution should be somewhere in-between thermal and Poissonian. However, here we conservatively assume thermal statistics, which leads to slightly larger values of $P_2$ and thus slightly lower values of the entanglement depth extracted from the measured data (Fig. 3).

Due to loss, a photon created in mode a will reach the detector $D_a$ with probability of $\eta_{a'} < 1$. It will be detected with a probability given by the detection efficiency $\eta_{D_a}$. We model the combined (limited) channel transmission and detector efficiency using a beam-splitter with transmission probability of $\eta_a = \eta_{a'} \times \eta_{D_a}$ followed by a perfect detector. Thus, the the creation operator for mode a transforms as:

$$a^\dagger \rightarrow \sqrt{\eta_a} a^\dagger + \sqrt{1 - \eta_a} a_l^\dagger, \quad (11)$$

where $a_l^\dagger$ is the creation operator in the loss mode of this hypothetical beam-splitter. With this transformation we can write:

$$\left(a^\dagger\right)^n = \sum_{k=0}^{n} \binom{n}{k} \left(\sqrt{\eta_a} a^\dagger\right)^k \left(\sqrt{1 - \eta_a} a_l^\dagger\right)^{n-k}. \quad (12)$$

The state of the loss mode $a_l$ and the mode b together, after a $k$-photon detection in the detector $D_a$, becomes:

$$|\overline{\psi}\rangle_k = \sum_{n=k}^{\infty} \sqrt{\binom{n}{k}} \xi_n \sqrt{\eta_a} k \sqrt{(1 - \eta_a)^{(n-k)}} |n - k\rangle_{a_l} |n\rangle_b. \quad (13)$$

Considering the fact that $D_a$ is a detector that does not resolve photon numbers, we sum over all possible values of $k \geq 1$ in mode a and trace out the loss channel from the state $|\overline{\psi}\rangle_k$. This results in the state of the mode b after heralding by $D_a$ as:

$$\rho_b = \frac{1}{\mathcal{N}} \sum_{k=1}^{\infty} \sum_{n=k}^{\infty} \binom{n}{k} |\xi_n|^2 \eta_a^k (1 - \eta_a)^{(n-k)} |n\rangle_b \langle n| \quad (14)$$

$$= \sum_{n=1}^{\infty} p_n |n\rangle_b \langle n|,$$

with

$$\mathcal{N} = \sum_{k=1}^{\infty} \sum_{n=k}^{\infty} \binom{n}{k} |\xi_n|^2 \eta_a^k (1 - \eta_a)^{(n-k)}, \quad (15)$$

$$p_n = \frac{\sum_{k=1}^{n} \binom{n}{k} |\xi_n|^2 \eta_a^k (1 - \eta_a)^{(n-k)}}{\mathcal{N}}. \quad (16)$$

In mode b, each photon experiences loss (before entering the AFC) that is modeled by a beam-splitter with transmission probability $\eta_b$. Each photon is then absorbed by the AFC with probability $\eta_w$. Otherwise, it is transmitted through the AFC and detected by the detector $D_b$, similarly modeled as a beam-splitter with transmission probability of $\eta_t$ followed by a perfect detector. Thus, the creation operator for mode b transforms as:

$$b^\dagger \rightarrow \sqrt{\eta_b \eta_w} b^\dagger + \sqrt{\eta_b (1 - \eta_w) \eta_t} b_t^\dagger$$

$$+ \sqrt{\eta_b (1 - \eta_w)(1 - \eta_t)} b_{tl}^\dagger + \sqrt{1 - \eta_b} b_l^\dagger, \quad (17)$$

where $b_t^\dagger$, $b_l^\dagger$, and $b_{tl}^\dagger$ are the creation operators in the transmission mode, loss

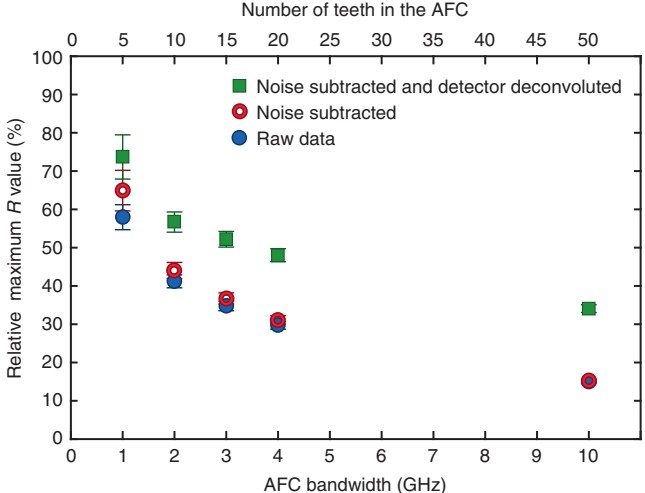

**Fig. 8** Echo contrast depends on atomic frequency comb bandwidth. This figure shows the echo contrast $R$ as a percentage of the total number of teeth as a function of the bandwidth of the atomic frequency comb. *Error bars* indicate standard deviations derived from the Gaussian fitting and Poissonian statistics of the photon detection events

mode before the AFC, and the loss mode after the AFC, respectively. Hence, the associated optical state becomes:

$$\rho_{\text{b}} = \sum_n p_n |\Phi_n\rangle \langle \Phi_n|, \tag{18}$$

where

$$
|\Phi_n\rangle = \frac{1}{\sqrt{n!}} \left( \sqrt{\eta_{\text{b}} \eta_{\text{w}}} b^\dagger + \sqrt{\eta_{\text{b}}(1-\eta_{\text{w}})\eta_{\text{t}}} b_{\text{t}}^\dagger \right.
$$
$$
\left. + \sqrt{\eta_{\text{b}}(1-\eta_{\text{w}})(1-\eta_{\text{t}})} b_{\text{tl}}^\dagger + \sqrt{1-\eta_{\text{b}}} b_{\text{l}}^\dagger \right)^n |0\rangle.
$$

Conditioning on "no detection" in the $b_{\text{t}}$ mode (since a detection would correspond to a photon that was not absorbed in the AFC) results in the density matrix

$$\tilde{\rho}_{\text{b}} = \frac{1}{\mathcal{M}} \sum_n \overline{p}_n |\tilde{\Phi}_n\rangle \langle \tilde{\Phi}_n|, \tag{19}$$

where

$$|\tilde{\Phi}_n\rangle = \frac{1}{(\eta_{\text{b}}\eta_{\text{w}} + Z^2)^{n/2}} |\overline{\Phi}_n\rangle,$$

$$|\overline{\Phi}_n\rangle = \frac{1}{\sqrt{n!}} \left( \sqrt{\eta_{\text{b}}\eta_{\text{w}}} b^\dagger + Zx^\dagger \right)^n |0\rangle, \tag{20}$$

$$\overline{p}_n = p_n (\eta_{\text{b}}\eta_{\text{w}} + Z^2)^n,$$

$$\mathcal{M} = \sum_n p_n (\eta_{\text{b}}\eta_{\text{w}} + Z^2)^n.$$

We have used

$$Z = \sqrt{\eta_{\text{b}}(1-\eta_{\text{w}})(1-\eta_{\text{t}}) + (1-\eta_{\text{b}})}, \tag{21}$$

$$x^\dagger = \frac{1}{Z} \left( \sqrt{\eta_{\text{b}}(1-\eta_{\text{w}})(1-\eta_{\text{t}})} b_{\text{tl}}^\dagger + \sqrt{1-\eta_{\text{b}}} b_{\text{l}}^\dagger \right).$$

We can rewrite the state $\tilde{\rho}_{\text{b}}$ in the number basis of the b mode and the x mode

(defined in the previous equation) as:

$$\tilde{\rho}_{\text{b}} = \frac{1}{\mathcal{M}} \left\{ \sum_{n=1}^\infty p_n \sum_{r,r'=1}^n \sqrt{\binom{n}{r}\binom{n}{r'}} \sqrt{\eta_{\text{b}}\eta_{\text{w}}}^{r+r'} Z^{2n-r-r'} \right.$$
$$\left. |r, n-r\rangle_{\text{b,x}} \langle r', n-r'| \right\}. \tag{22}$$

The probability $P_r$ that $r$ photons are present in the AFC is calculated as:

$$P_r = {}_{\text{b}}\langle r| \text{Tr}_x \tilde{\rho}_{\text{b}} |r\rangle_{\text{b}} = \frac{1}{\mathcal{M}} \sum_{n=r}^\infty p_n \binom{n}{r} (\eta_{\text{b}}\eta_{\text{w}})^r Z^{2n-2r} \tag{23}$$

Therefore, we can obtain the values of $P_1$ and $P_2$ by measuring $\mu$, $\eta_{\text{a}}$, $\eta_{\text{b}}$, $\eta_{\text{w}}$, and $\eta_{\text{t}}$.

**Experimental estimation of $\mu$.** The second-order cross-correlation function $g(\tau)_{\text{ab}}^{(2)}$ gives information about the photon number distribution of a two-mode field. Specifically, for the downconverted photons in mode a (1532 nm) and mode b (795 nm) at zero time delay ($\tau = 0$), it can be written as:

$$g_{\text{ab}}^{(2)}(0) = \frac{p_{\text{ab}}(0)}{p_{\text{a}} p_{\text{b}}}, \tag{24}$$

where $p_{\text{a}}$ ($p_{\text{b}}$) is the probability for a detection in mode a (b) and $p_{\text{ab}}(0)$ the probability to detect a coincidence in a temporal window centered at $\tau = 0$. Experimentally, we obtained $g_{\text{ab}}^{(2)}$ as the ratio between the coincidence rate $C_{\text{ab}}$ at $\tau = 0$ (where photons in mode a and b exhibit maximum correlations) and the coincidence rate at a delay $\tau$ larger than the coherence length of the photons in mode a and b (where photon creation, and hence detection, in mode a and b is completely independent).

All experiments were performed at maximum pump power of $\approx 500\,\mu\text{W}$, for which we measured $g_{\text{ab}}^{(2)}(0) = 884 \pm 50$. In the limit where $g_{\text{ab}} \gg 1$, the cross-correlation function can be written as $g_{\text{ab}}(0) = \frac{1.44}{\mu}$. This allows us to estimate a value of $\mu = (1.1 \pm 0.1) \times 10^{-3}$. Note that the uncertainty in $\mu$ is the dominant contribution to the uncertainty of $P_2$ given in the manuscript. Our coincidence measurements were performed using home-made logic electronics with a detection window of 5 ns. A characterization of $g_{\text{ab}}^{(2)}$ as a function of the pump power can be seen in Fig. 2.

**Experimental estimation of $\eta_{\text{a}}$, $\eta_{\text{b}}$, and $\eta_{\text{t}}$.** To estimate $\eta_{\text{a}}$ and $\eta_{\text{b}}$, we change the set-up represented in Fig. 6 and place the detector $D_{\text{b}}$ (with detection efficiency of $\eta_{D_{\text{b}}}$) before the AFC, i.e., the photons in mode b only pass through the loss channel with probability $\eta_{\text{b}^*}$ before they reach $D_{\text{b}}$. In this configuration, both $\eta_{\text{a}}$ and $\eta_{\text{b}^*}$ can be estimated from the coincidence detection rate $C_{\text{ab}}$ and the single detection rates $S_{\text{a}}$ and $S_{\text{b}}$. Since $\mu \ll 1$ in our case, we can write:

$$
\begin{aligned}
C_{\text{ab}} &= \mu \eta_{\text{a}^*} \eta_{D_{\text{a}}} \eta_{\text{b}^*} \eta_{D_{\text{b}}} / \tau_{\text{P}}, \\
S_{\text{a}} &= \mu \eta_{\text{a}^*} \eta_{D_{\text{a}}} / \tau_{\text{P}}, \\
S_{\text{b}} &= \mu \eta_{\text{b}^*} \eta_{D_{\text{b}}} / \tau_{\text{P}},
\end{aligned}
\tag{25}
$$

where $\tau_{\text{P}}$ is the coherence time of the pump laser. From the above relations, we find:

$$
\begin{aligned}
\eta_{\text{a}} &= \frac{C_{\text{ab}}}{S_{\text{b}}}, \\
\eta_{\text{b}^*} &= \frac{C_{\text{ab}}}{S_{\text{a}} \eta_{D_{\text{b}}}}.
\end{aligned}
\tag{26}
$$

We experimentally find $\eta_{\text{a}} = 11.0\%$ and $\eta_{\text{b}^*} = 5.3\%$, for which we used $\eta_{D_{\text{b}}} = 0.60$. In the set-up depicted in Fig. 6, we can write $\eta_{\text{b}} = \eta_{\text{b}^*} \times \eta_{c_i}$, where $\eta_{c_i}$ is the probability with which a photon that passes through the loss will enter the AFC. Note that the background absorption of the AFC $d_0$, as shown in Fig. 7, can be treated as loss and is hence included in $\eta_{c_i}$. Since $d_0$ varies for AFCs with different $N$, $\eta_{c_i}$ and hence $\eta_{\text{b}}$ also varies. For instance, the AFC with $N = 564$ and $B = 6$ GHz for which entanglement depth is calculated in Fig. 3b, results in $\eta_{\text{b}} = 1.06\%$. For the two AFCs shown in Fig. 4 with $N = 9$ and bandwidths of $B = 6$ GHz and $B = 0.33$ GHz, we estimate $\eta_{\text{b}} = 1.99\%$ and $\eta_{\text{b}} = 0.96\%$, respectively.

The probability with which a photon that is transmitted through the AFC is detected by the detector $D_{\text{b}}$, can be written as $\eta_{\text{t}} = \eta_{\text{t}^*} \times \eta_{D_{\text{b}}}$. Here, $\eta_{\text{t}^*}$ is the probability that the photon passes through the loss channel after the AFC, reaches into the detector $D_{\text{b}}$ which has a detection efficiency of $\eta_{D_{\text{b}}}$. We estimate $\eta_{\text{t}} = 36.0\%$.

**Experimental estimation of AFC write efficiency $\eta_w$.** The write efficiency of the AFC is the probability for the AFC to absorb a photon. It can be calculated from the effective optical depth of the AFC as:

$$\eta_w = 1 - e^{-\frac{d_1}{F}}, \tag{27}$$

where $d_1$ is the peak absorption, $F$ is the finesse of the AFC, and $d_1/F$ is the effective optical depth of the comb (see Fig. 7). For the different AFCs that we create, $d_1$ varies, and hence the write efficiency $\eta_w$ also varies. For instance, the write efficiencies for the broadband AFCs with number of teeth $N = 564$ and $N = 9$ are 33% and 54%, respectively. For the narrowband AFC with $N = 9$, which is shown in Fig. 4, we estimate $\eta_w = 0.9\%$.

The dominant contribution to the uncertainty in $P_1$ given in the manuscript comes from the uncertainty in the optical depth of around 10%. The uncertainties associated with the measurements of the other efficiencies are negligible in comparison.

**Heralded autocorrelation $g_{bb}^{(2)}(0)$.** The second-order zero-time autocorrelation function $g_{bb}^{(2)}(0)$ is a witness of non-classicality for single-mode fields. To measure $g_{bb}^{(2)}(0)$, we add a balanced (50/50) beam-splitter (BS) before the detection of the photon in mode b (795 nm photon), and measure the probability to detect a coincidence between the two BS outputs ($p_{b_1 b_2}$), as well as the single output probabilities ($p_{b_1}$ and $p_{b_2}$), all conditioned on the detection of a photon in mode a (1532 nm photon). We can now write the autocorrelation function $g_{bb}^{(2)}$ as:

$$g_{bb}^{(2)}(0) = \frac{p_{b_1 b_2}(0)}{p_{b_1} p_{b_2}}. \tag{28}$$

For a perfect single-photon state $g_{bb}^{(2)}(0) = 0$, whereas for a coherent state $g_{bb}^{(2)}(0) = 1$. We find $g_{bb}^{(2)}(0) = 0.0024 \pm 0.0006$, see Fig. 2b, proving the nearly ideal nature of our heralded single-photon source. This value is consistent with the value of $\mu$ determined from measuring the cross-correlation function above[45].

**Background noise estimation and subtraction.** The collection of our experimental data suffers from imperfections that limit the the values of $R$, but are unrelated to the detection of the heralded single photon. In our analysis, we account for two such sources of imperfection, namely background noise and limited temporal resolution of the detector, where the latter is treated in the next section. The background noise causes a constant level of detection events, which we can assess by averaging over the detection events before a heralding photon detection as indicated by the *red region* in the *insert* of Fig. 3. The background noise is constant at about 0.9 counts per 80 ps data bin for a 15 min measurement. Since this background noise is completely uncorrelated with the heralded photon, we can subtract it in order to obtain a more accurate echo contrast. Clearly, the impact of such noise subtraction will increase when the echo signal is weakened, e.g., due to increased AFC storage times or reduction of the AFC bandwidth, as evidenced by the data shown in Figs. 3, 4, and 8.

We can determine the origin of the background noise from a series of independent measurements. Detector dark counts contribute about 12%, while noise due to leaked optical pumping light and spontaneous emission from atoms excited during optical pumping amounts to ~11%. Finally, the photon pair source may generate additional pairs other than that causing the heralding event, as discussed above in the context of the second-order cross-correlation function. If the 795 nm photon member of one of these additional pairs reaches the detector after the AFC, it will appear as a background noise count. This is the main contribution to the background noise at around 77%. The relative contributions to the background noise depend slightly on the AFC efficiency and loss.

**Deconvolution of the detector response.** The temporal shape of the heralded photon is given by the Fourier transform of its spectral distribution. At the AFC input, the spectrum is defined by the 6 GHz FP filter, whereas after re-emission from the AFC the photon spectrum will depend on the overall bandwidth as well as exact shape of the AFC. However, when the heralded photon is detected, the inherent detector jitter can smear out its temporal shape thus reducing the signal-to-noise ratio and thereby the echo contrast.

The joint detector response is measured by removing all spectral filtering elements and recording the distribution of correlated detection immediately after the, now extremely broadband, pair source. We find that the joint detector response is well approximated by a Gaussian function of full-width-at-half-maximum (FWHM) $\Delta t_D = 354$ ps. Fitting the echoes with a Gaussian function with amplitude $E$ and a FWHM of $\Delta t_p$ allows us to deconvolute the detector response and compute a corrected signal amplitude of

$E' = E \sqrt{\Delta t_p^2 / \left( \Delta t_p^2 - \Delta t_D^2 \right)}$. Note that for the broadband AFCs, the output photon duration is on the order of $\Delta t'_p \sim 100 - 200$ ps, which is significantly smeared by the detector jitter. As expected, the deconvolution of the detector response leads to a large increase of the echo contrast for the broadband AFCs and

less of an increase for narrowband AFCs, for which the temporal duration of the echoes exceeds the detector jitter—this interplay is evident in Figs. 4 and 8.

**Number of atoms corresponding to a single AFC tooth.** We calculate the number of atoms that correspond to a single AFC tooth using two complementary approaches. The first method utilizes the experimentally determined absorption spectrum and the Tm-atom density of the Tm:LiNbO₃ crystal, while the second method relies on single-ion spectroscopic properties.

For the first method, we note that the integrated absorption spectrum $\Theta$ of an inhomogeneously broadened transition of an atomic ensemble is $\Theta_i = \int \alpha(\nu) L d\nu$, where $\alpha(\nu)$ is the absorption coefficient resulting from all transitions that feature a resonance frequency $\nu$, and $L$ is the length of the medium. Similarly, the integrated absorption spectrum of a single AFC tooth is $\Theta_t$, where the integration is taken over the tooth. If a laser beam of cross-sectional area $A$ (defined by FWHM of the intensity distribution of a Gaussian beam) is sent through a medium that features atom density $n_d$, then the number of atoms within the beam is $n_d L A$, and the number of atoms corresponding to a single tooth is $N_t^{(1)} = n_d L A (\Theta_t / \Theta_i)$.

For the second method, we calculate the optical depth $d_{atom}$ that corresponds to a single atom in a crystal using $d_{atom} = [(n^2 + 2)^2 / (72\pi n A \sigma^2)](\gamma_s / \Gamma_h)$, where $\gamma_s$ and $\Gamma_h$ are the spontaneous emission rate and homogeneous broadening of the transition, respectively, $n$ is the index of refraction, and $\sigma = \nu/c$[46]. Since $d_{atom}$ refers to an atom that features a linewidth of $\Gamma_h$, we estimate the number of atoms that correspond to a single tooth using $N_t^{(2)} = \Theta_t / (\Gamma_h d_{atom})$.

For the ³H₆ → ³H₄ transition of Tm:LiNbO₃, the Tm-atom number density is $n_d = 1.89 \times 10^{19}$ cm⁻³, $n = 2.256$, $\Gamma_h = 10$ kHz, $\gamma_s = 2.6$ kHz, and $\int \alpha(\sigma) d\sigma = 497$ cm⁻²[47], where $\Theta_i = Lc \int \alpha(\sigma) d\sigma$. The laser beam that is used to determine the absorption spectra is collimated and has a cross-section of $A = \pi (80 \,\mu\text{m})^2$, and the crystal length is $L = 6.8$ mm. For our AFC with $N = 564$ teeth, we measure $\Theta_t = 4.3$ MHz, giving $N_t^{(1)} = 1.1 \times 10^9$ and $N_t^{(2)} = 1.7 \times 10^9$ atoms per tooth. The difference between the two estimates may be due to the measurement uncertainty or to the fact that not all transitions that contribute to the absorption line have identical properties.

**Echo contrast $R$ as a function of AFC bandwidth.** In order to create a perfect W state, the AFC structure should be uniformly illuminated by the incoming photon. This guarantees that each AFC tooth has the same probability to contribute to the photon absorption. In the experiment, we use a 6 GHz FWHM bandwidth photon with a Lorentzian profile given by the transmission profile of the used Fabry–Perot filtering cavity (FP). The uniformity of the absorption over the different teeth should thus depend on the bandwidth of the AFC. In order to analyze this effect, we measure, for a fixed storage time (5 ns), the echo contrast $R$ as a function of the prepared AFC bandwidth (see Fig. 8). Here $R$ is expressed in terms of the percentage with respect to the maximum attainable $R$ value given the known number of teeth created in each AFC. We observe that, as we narrow the AFC bandwidth, the relative $R$ value increases. This is consistent with the expectation that we are approaching the creation of an ideal W state. Note that our derived bounds on the entanglement depth are still correct, albeit too conservative, for the case where the created state is not a perfect W state, since a state with unequal coefficients has to involve more entangled teeth in order to generate the same value of $R$.

**Data availability.** Experimental data that support the findings of this study have been deposited at Open Science Framework (OSF) with the accession codes "JZMCA (https://osf.io/jzmca/), DOI 10.17605/OSF.IO/JZMCA".

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

## Acknowledgements

We thank F. Fröwis and F. Bussières for very useful discussions and V. Kiselyov for technical support. This work was funded through NSERC, AITF and the DARPA Quiness program subaward contract number SP0020412-PROJ0005188, under prime contract number W31P4Q-13-1-0004. V.B.V. and S.W.N. acknowledge partial funding for detector development from the Defense Advanced Research Projects Agency (DARPA) Information in a Photon (InPho) program. Part of the detector research was carried out at the Jet Propulsion Laboratory, California Institute of Technology, under a contract with the National Aeronautics and Space Administration. W.T. acknowledges funding as a Senior Fellow of the Canadian Institute for Advanced Research.

## Author contributions

N.S. and C.S. conceived the project with some input from W.T. The theoretical approach was developed by P.Z., S.K.G., K.H., and C.S., and the calculations performed by P.Z. and S.K.G. with guidance from C.S. The experiments were developed and performed by C.D., N.S., G.H.A., P.L., M.G., and D.O. with guidance from W.T. The detectors were designed and fabricated by V.B.V., F.M., M.D.S., and S.W.N. The paper was written by P.Z., C.D., P.L., G.H.A., S.K.G., M.G.P., D.O., N.S., W.T., and C.S.

## Additional information

**Competing interests:** The authors declare no competing financial interests.

