## [Peer Review File · Nature Communications]

Reviewers' Comments:

Reviewer #1:

Remarks to the Author:

It is well known that when a single photon is stored in an ensemble based quantum memory, the memory will be in a W state with one atom in the excited state and all others in the ensemble in the ground state. Hence the ensemble can be considered in a highly entangled state. What the authors have shown is that the maximum depth of entanglement of a quantum memory that can be observed by solely making a temporal/ frequency measurement of the output photons is given by the time bandwidth product of the memory. That is the ratio of the duration of the longest photon packet that the memory can store divided by the duration of the shortest. In the case of a AFC this is given by the number of teeth in the comb. This result is easy to see if you think of measuring the frequency of the photon packets readout from the memory. The number of distinguishable frequency windows (the number slits if we are drawing the analogy to a spatial grating) is given by the time bandwidth product. The authors also show that level of entanglement will be reduced if the single photon source is not ideal. Both these results seem obvious and no concrete case is made for their significance.

The depth of entanglement in a AFC quantum memory is then measured. It is hard to see how this result could not have been deduced from the authors previous demonstrations of AFC memories showing the storage of single photons.

I do not recommend this paper for publication as it does not contain new results of sufficient significance to warrant publishing in Nature Communications.

Reviewer #3:

Remarks to the Author:

Since the authors did only minor modifications and reshuffling of the former manuscript text and methods section, my opinion since the last review has not changed.

Although this is not 'the absolute breakthrough' result I can support the publication in Nature Communication.

Reviewer #4:

Remarks to the Author:

Authors detect multipartite entanglement in solid state systems. With a single excitation, they create a W-state and detect it as multipartite entangled.

In particular, they create an atomic frequency comb and detect entanglement between the teeth of the comb. The rare earth doped crystal absorbs a heralded single photon, then it emits it. They detect the photon and use the probability of having 1 and 2 photons, together with the echo contrast given in Eq. (2) to detect multipartite entanglement. They find genuine multipartite entanglement between 200 teeth of the comb. Every tooth behaves like a single qubit. The probabilities obtained are

$p_1 \sim 10^{-3}$
 $p_2 \sim 10^{-8}$

We see that $1 \gg p_1 \gg p_2$. If p_2 is small then the entanglement depth $M=R$ [Eq. (3)], where R is the echo contrast. I find the derivation of the criterion very interesting, and it is also very nice that R , a measurable quantity, is directly related to the entanglement depth.

I find the paper very well written, very clear. I think, the paper describes an astonishing experimental success. I suggest its publication in Nature Communications.

I would like to react to the comments of the other referees.

I would now mostly comment concerning entanglement detection.

While the findings of the paper are relevant to applications, I would like to argue that multipartite entanglement in solids is in itself of a large importance.

— Why solid state systems are important: There has been a large effort to detect multipartite entanglement of many particles in quantum systems in large ensembles. These involved so far almost exclusively cold gases.

There has not been results in condensed matter systems. The reason is that in condensed matter systems there are many noise sources that do not exist in cold atoms. However, for the future of quantum information science, it is crucial to realize quantum information processing in condensed matter.

Just for this reason, the results of the paper are outstanding.

-- Why the detection of entanglement depth is important: most quantum systems can be considered to be in a product state of few particle states. Such as

$\Phi_1 \otimes \Phi_2 \otimes \Phi_3 \dots$

where Φ_n are states of at most k atoms. One can also consider mixture of such states, which are just the quantum states with an entanglement depth at most k .

Such states naturally appear, for example, in quantum states in thermal equilibrium that can very well be described by an ansatz

$\rho_1 \otimes \rho_2 \otimes \rho_3$

where ρ_k are units of at most k qubits. As the temperature increases, the minimal k that gives a good enough description of the state decreases.

Hence, the quantum system appears as being made of such groups of atoms that do not interact with each other.

Thus, in theory, the $T=0$ ground state of many spin models possess genuine multipartite entanglement, in practice, when such states are realized at finite temperatures and in a noisy environment, we can obtain quantum states that can be described by little particles groups among which there are only classical correlations.

Hence, the question arises: is it possible to have large scale entanglement in a quantum system? The answer could well be no. It could happen that they cannot create more than, say, 10 particle

entanglement due to various noise effects.

On the other hand, the possibility of having large entanglement depth and large scale multipartite entanglement is necessary for real quantum information processing applications. And it is also a very important to prove that multipartite entanglement can be created from fundamental physics point of view.

Concerning future work, I would like to make a comment. One could try to find systems in which there is entanglement between spatially separated parties. Perhaps, this would be possible with some modification of the setup, for example, with two crystals. A proof of principle demonstration of entanglement between spatially separated parties could be very important and very interesting.

Response to referees

Below we give our detailed response to the reviewers' comments.

Reviewer #1 (Remarks to the Author):

It is well known that when a single photon is stored in an ensemble based quantum memory, the memory will be in a W state with one atom in the excited state and all others in the ensemble in the ground state. Hence the ensemble can be considered in a highly entangled state. What the authors have shown is that the maximum depth of entanglement of a quantum memory that can be observed by solely making a temporal/ frequency measurement of the output photons is given by the time bandwidth product of the memory. That is the ratio of the duration of the longest photon packet that the memory can store divided by the duration of the shortest. In the case of a AFC this is given by the number of teeth in the comb. This result is easy to see if you think of measuring the frequency of the photon packets readout from the memory. The number of distinguishable frequency windows (the number slits if we are drawing the analogy to a spatial grating) is given by the time bandwidth product. The authors also show that level of entanglement will be reduced if the single photon source is not ideal. Both these results seem obvious and no concrete case is made for their significance.

The depth of entanglement in a AFC quantum memory is then measured. It is hard to see how this result could not have been deduced from the authors previous demonstrations of AFC memories showing the storage of single photons.

I do not recommend this paper for publication as it does not contain new results of sufficient significance to warrant publishing in Nature Communications.

We disagree with the referee's claim that our result on the entanglement depth is 'easy to see'. The derivation of a precise mathematical bound for the entanglement depth was far from trivial and is one of the main results of our work. The referee makes no mention of the fact that the entanglement that is present in the system essentially depends on the statistics of the light that is stored. However, this is the key question that is resolved by our bound. We now highlight this point in the introduction.

Regarding the referee's suggestion that our results could have been deduced from previous experiments, we already addressed this point in our previous response. Previous experiments did not measure all the relevant parameters with sufficient precision, which is understandable, given that the bound for the entanglement depth was not known.

Reviewer #3 (Remarks to the Author):

Since the authors did only minor modifications and reshuffling of the former manuscript text and methods section, my opinion since the last review has not changed.

Although this is not 'the absolute breakthrough' result I can support the publication in Nature Communication.

We thank the referee for his or her positive recommendation.

Reviewer #4 (Remarks to the Author):

Authors detect multipartite entanglement in solid state systems. With a single excitation, they create a W-state and detect it as multipartite entangled.

In particular, they create an atomic frequency comb and detect entanglement between the teeth of the comb. The rare earth doped crystal absorbs a heralded single photon, then it emits it. They detect the photon and use the probability of having 1 and 2 photons, together with the echo contrast given in Eq. (2) to detect multipartite entanglement. They find genuine multipartite entanglement between 200 teeth of the comb. Every tooth behaves like a single qubit. The probabilities obtained are

$p_1 \sim 10^{-3}$

$p_2 \sim 10^{-8}$

We see that $1 \gg p_1 \gg p_2$. If p_2 is small then the entanglement depth $M=R$ [Eq. (3)], where R is the echo contrast. I find the derivation of the criterion very interesting, and it is also very nice that R , a measurable quantity, is directly related to the entanglement depth.

I find the paper very well written, very clear. I think, the paper describes an astonishing experimental success. I suggest its publication in Nature Communications.

We are grateful to the referee for this assessment.

I would like to react to the comments of the other referees.

I would now mostly comment concerning entanglement detection.

While the findings of the paper are relevant to applications, I would like to argue that multipartite entanglement in solids is in itself of a large importance.

— Why solid state systems are important: There has been a large effort to detect multipartite entanglement of many particles in quantum systems in large ensembles. These involved so far almost exclusively cold gases.

There has not been results in condensed matter systems. The reason is that in condensed matter systems there are many noise sources that do not exist in cold atoms. However, for the future of quantum information science, it is crucial to realize quantum information processing in condensed matter.

We thank the referee for clearly articulating these points. We completely agree.

Just for this reason, the results of the paper are outstanding.

-- Why the detection of entanglement depth is important: most quantum systems can be considered to be in a product state of few particle states. Such as

$\Phi_1 \otimes \Phi_2 \otimes \Phi_3 \dots$

where Φ_n are states of at most k atoms. One can also consider mixture of such states, which are just the

quantum states with an entanglement depth at most k .

Such states naturally appear, for example, in quantum states in thermal equilibrium that can very well be described by an ansatz

$\rho_1 \otimes \rho_2 \otimes \rho_3$

where ρ_k are units of at most k qubits. As the temperature increases, the minimal k that gives a good enough description of the state decreases.

Hence, the quantum system appears as being made of such groups of atoms that do not interact with each other.

Thus, in theory, the $T=0$ ground state of many spin models possess genuine multipartite entanglement, in practice, when such states are realized at finite temperatures and in a noisy environment, we can obtain quantum states that can be described by little particles groups among which there are only classical correlations.

Hence, the question arises: is it possible to have large scale entanglement in a quantum system? The answer could well be no. It could happen that they cannot create more than, say, 10 particle entanglement due to various noise effects.

Again we thank the referee for clearly making this point. We have made an effort to state this more clearly in the new version of the introduction.

On the other hand, the possibility of having large entanglement depth and large scale multipartite entanglement is necessary for real quantum information processing applications. And it is also very important to prove that multipartite entanglement can be created from fundamental physics point of view.

We agree.

Concerning future work, I would like to make a comment. One could try to find systems in which there is entanglement between spatially separated parties. Perhaps, this would be possible with some modification of the setup, for example, with two crystals. A proof of principle demonstration of entanglement between spatially separated parties could be very important and very interesting.

We thank the referee for pointing this out. We mention this interesting possibility in the discussion section.